# A dendritic guidance receptor functions in both ligand dependent and independent modes

**Anay R. Reddy**[1], **Sebastian J. Machera**[2,3,4], **Zoe T. Cook**[1,5], **Huichao Deng**[1], **Wioletta I. Nawrocka**[2,3,4], **Engin Özkan**[2,3,4], **Kang Shen**[1,6*]

**1** Department of Biology, Stanford University, Stanford, California, United States of America,
**2** Department of Biochemistry and Molecular Biology, The University of Chicago, Chicago, Illinois, United States of America, **3** Institute for Neuroscience, The University of Chicago, Chicago, Illinois, United States of America, **4** Institute for Biophysical Dynamics, The University of Chicago, Chicago, Illinois, United States of America, **5** Neurosciences IDP, Stanford University, Stanford, California, United States of America, **6** Howard Hughes Medical Institute, Stanford University, Stanford, California, United States of America

* kangshen@stanford.edu

## Abstract

The formation of an appropriately shaped dendritic arbor is critical for a neuron to receive information. Dendritic morphogenesis is a dynamic process involving growth, branching, and retraction. How the growth and stabilization of dendrites are coordinated at the molecular level remains a key question in developmental neurobiology. The highly arborized and stereotyped dendritic arbors of the *Caenorhabditis elegans* PVD neuron are shaped by the transmembrane DMA-1 receptor through its interaction with a tripartite ligand complex consisting of SAX-7/L1CAM, MNR-1/FAM151B, and LECT-2/LECT2. However, receptor null mutants exhibit strongly reduced dendrite outgrowth, whereas ligand null mutants show disordered branch patterns, suggesting a ligand-independent function of the receptor. To test this idea, we identified point mutations in *dma-1* that disrupt receptor-ligand binding and introduced corresponding mutations into the endogenous gene. We show that the ligand-free receptor is sufficient to drive robust, disordered dendritic branch formation but results in a complete loss of arbor shape. This disordered outgrowth program utilizes similar downstream effectors as the stereotyped outgrowth program, further arguing that ligand binding is not necessary for outgrowth. Finally, we demonstrate that ligand binding is required to maintain higher-order dendrites after development is complete. Taken together, our findings support a surprising model in which ligand-free and ligand-bound DMA-1 receptors have distinct functions: the ligand-free receptor promotes stochastic outgrowth and branching, whereas the ligand-bound receptor guides stereotyped dendrite morphology by stabilizing arbors at target locations.

**Data availability statement:** All quantifications, image files, and raw data generated for this manuscript are available at https://github.com/anayreddy1/Reddyetal2025.git.

**Funding:** K. Shen is an investigator in the Howard Hughes Medical Institute. This work was funded by CMB Training Grant T32 GM007276 to AR and an NIH RO1 NS082208 awarded to KS. The funders had no role in study design, data collection and analysis, decision to publish, or preparation of the manuscript.

**Competing interests:** The authors have declared that no competing interests exist.

## Author summary

Formation of a dendritic arbor requires the balancing between outgrowth and stabilization. Here, we aim to understand how this coordination is achieved at the molecular level. Stereotyped arborization of the *C. elegans* PVD neuron requires a tripartite ligand complex, the DMA-1 receptor, and actin regulators, leading to a model in which ligand-receptor binding is necessary for outgrowth. However, it is less clear how ligand-receptor interaction regulates the growth and stabilization of dendrites. Genetic evidence demonstrates that while the DMA-1 receptor and actin regulators are critical for dendrite outgrowth, the tripartite ligand complex is dispensable for outgrowth but required for dendrite shape. In this work, we show that the disruption of ligand-receptor interaction leads to disordered dendritic arborization, demonstrating that ligand-receptor binding is not required for outgrowth. This disorganized arborization requires similar actin regulators that are employed in stereotyped arborization, arguing that activation of downstream effectors can occur independently of ligand binding. Finally, we demonstrate that ligand-receptor binding is required for stabilization of dendrites. Taken together, our work supports a model in which the DMA-1 receptor balances dendrite outgrowth and stabilization by functioning in a ligand-independent mode to promote stochastic outgrowth and a ligand-dependent mode to promote stabilization.

## Introduction

A neuron's ability to receive synaptic inputs or sensory stimuli is dependent on dendritic morphology. Size, shape, and branching pattern are key parameters of dendrite arbors that vary widely across neuronal cell types. During development, dendrite morphogenesis requires the integration of extrinsic signals derived from the nearby complex tissue environment, which ultimately instructs dendrite shape, with intrinsic cytoskeletal rearrangements to drive dendrite outgrowth [1]. A critical question in developmental neurobiology is how extrinsic and intrinsic mechanisms converge to form and maintain stereotyped dendritic arbors.

Guidance cues serve as extrinsic signals to instruct the growth direction of major dendritic processes during dendrite morphogenesis in a manner that resembles axon development. For example, a putative gradient of the Sema3A ligand near the marginal zone attracts apical dendrites of hippocampal CA1 pyramidal neurons towards the pial surface in mice [2]. Similarly, high levels of the Slit ligand at the central nervous system midline in *Drosophila melanogaster* repels motoneuron dendrites [3]. The Sema3A and Slit ligands exert their instructive roles on dendrite arborization through binding their cognate receptors neuropilin-1 and Robo, respectively [2,4]. Similar findings studying other ligand-receptor systems, such as Netrin/Frazzled and Wnt/Frizzled, supported a model in which cognate guidance receptors are activated by ligand binding, thereby triggering downstream intracellular effectors to regulate

dendrite outgrowth [1,5]. These examples suggest that diffusible gradients of attractants and repellants guide the morphology of major dendritic branches.

Additionally, fine dendritic branch morphologies must be continuously refined to establish a non-overlapping dendritic field both within the same neuron and between the neurons of a similar class. Self-avoidance and dendritic tiling mechanisms play important roles to minimize overlap [6]. These growth repulsive mechanisms ensure maximum coverage of a target field while minimizing overlap between branches. Repulsion between branches is achieved through *cis* and *trans* homophilic interactions between alternatively-spliced cell adhesion molecules, including DSCAM and protocadherins [7–9]. While the mechanisms regulating spacing between fine dendritic branches are well-understood, the molecular mechanisms that drive pathfinding of fine dendritic branches remain unclear. Furthermore, how receptors regulate the cytoskeletal machinery to refine the shape of fine branches is not fully understood.

Guidance receptors regulate dendrite outgrowth through the recruitment and activation of cytoskeletal regulators. For example, receptor-mediated dendrite outgrowth can employ the Rho GTPases RhoA, Rac1, and Cdc42 [10,11].These molecules cycle between an inactive GDP-bound state and an active GTP-bound state [12]. In the GTP-bound active state, Rho GTPases promote actin nucleation and polymerization to drive outgrowth [13,14]. Transitioning from the inactive conformation to the active state is accelerated by guanine nucleotide exchange factors (GEFs), which facilitate the replacement of GDP with GTP [12]. Guidance receptors control actin dynamics by recruiting GEFs. For example, upon binding to an ephrinB ligand, the EphB2 receptor phosphorylates and recruits the Rac GEF Tiam1, which promotes dendritic spine development [15]. Therefore, receptors can bridge extrinsic cues with intrinsic outgrowth programs by recruiting cytoskeletal regulators.

The PVD mechanosensory neuron in the nematode *Caenorhabditis elegans* (*C. elegans*) affords a powerful experimental system to study dendrite morphogenesis and dendrite branch patterning. PVD has an expansive dendritic arbor with orthogonally oriented primary, secondary, tertiary, and quaternary dendrites [16] (Fig 1a). The higher-order secondary, tertiary, and quaternary branches form structures resembling menorahs [16,17]. Menorah formation requires three extrinsic cues: the cell adhesion molecules SAX-7/L1CAM and MNR-1/FAM151B expressed by the hypodermis, and the chemotaxin LECT-2/LECT2 secreted from the muscle cells [18–22]. All three ligands collectively form a tripartite ligand complex and bind to the extracellular domain of the DMA-1 receptor expressed by PVD [21,22]. The intracellular domain of DMA-1 promotes dendrite outgrowth by recruiting the Rac GEF TIAM-1/TIAM1 [18,23]. DMA-1 also physically interacts with the claudin-like co-receptor HPO-30, which recruits the pentameric Wave Regulatory Complex (WRC) through its cytoplasmic tail [18,23,24]. Together, TIAM-1/TIAM1 and WRC activate Arp2/3 mediated actin assembly to drive dendrite outgrowth [25]. These findings are consistent with a model in which DMA-1 bridges extrinsic and intrinsic programs by promoting cytoskeletal changes in a manner dependent upon ligand binding (Fig 1b). It is conceivable that the tripartite ligand complex localizes to specific tissues to guide a deterministic outgrowth program which leads to a stereotyped dendritic arbor. In support of this model, both ligand null and receptor null mutants completely lack menorahs. However, ligand null mutants display robust but disordered branches, whereas receptor null mutants have sparse higher-order branching, arguing that the receptor can trigger dendrite outgrowth independently of ligand binding [21,22,26]. Furthermore, time-lapse microscopy of developing PVD neurons reveals that some secondary dendrite outgrowth events are stochastic in direction and can retract back to the primary dendrite. These findings demonstrate that PVD dendrite development is not fully deterministic and challenge the canonical ligand-receptor model in PVD dendrite morphogenesis [17].

To further understand the relationship between ligand-receptor interaction and dendrite development, we generated a mutant receptor incapable of binding ligand to test whether dendrite outgrowth of a highly organized arbor can occur in the absence of ligand-receptor binding. We employed an *in silico* approach to design what we term a "non-binding" mutant receptor and validated that ligand-receptor binding is abrogated through a biochemical approach. Genetic experiments reveal that the non-binding receptor is sufficient to drive disordered dendrite formation that phenocopies ligand null conditions. The disordered outgrowth phenotypes in this mutant require the same downstream effectors utilized in stereotyped

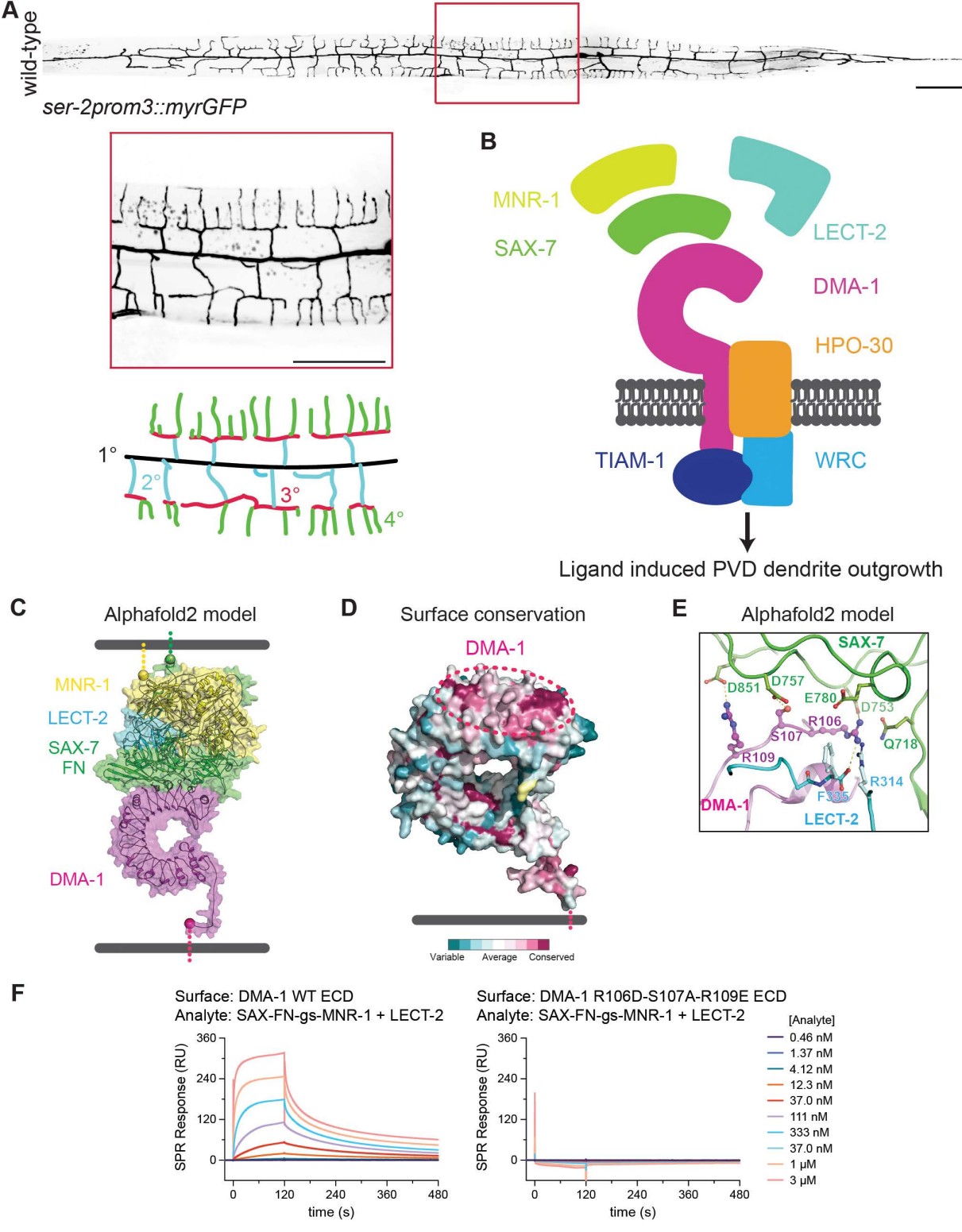

**Fig 1. Three receptor residues are necessary for DMA-1 receptor binding to a tripartite ligand complex. (A)** (Top) Fluorescence z-projections of the entire PVD neuron visualized with a *ser-2prom3::myristoylated::GFP* membrane marker in a young adult animal. Red box indicates region 150

μm anterior of PVD cell body. Scale bar, 100μm. (Middle) Inset enlargement of boxed region. Scale bar, 50 μm. (Bottom) Schematic of PVD dendrite branching. Primary (1°) dendrite shown in black, secondary (2°) dendrites shown in blue, tertiary (3°) dendrites shown in red, quaternary (4°) dendrites shown in green. (**B**) Original model depicting ligand-receptor binding bridging extrinsic signals with intrinsic cytoskeletal changes to activate PVD dendrite outgrowth. In this model, DMA-1 receptor binds an extracellular tripartite ligand complex consisting of SAX-7/L1CAM, LECT-2/LECT2, and MNR-1/FAM151B, which triggers activation of actin polymerization through the downstream Rac GEF TIAM-1/TIAM1, coreceptor HPO-30, and Wave Regulatory Complex (WRC). (**C**) Alphafold 2 (multimer) model of DMA-1 ectodomain + LECT-2/LECT2 + SAX-7/L1CAM FN domains + MNR-1/FAM151B. The Alphafold2 model orients the C-terminal ends of MNR-1/FAM151B and SAX-7/L1CAM at the expected site for the skin cell membrane. (**D**) Surface conservation of DMA-1 highlights the ligand engagement surface as highly conserved (dashed oval). (**E**) Close-up of residues DMA-1 R106 to R109 in the Alphafold 2 model. (**F**) SPR sensorgrams for the binding of the tripartite ligand complex on wild-type and mutant DMA-1 ectodomain (ECD).

outgrowth of organized arbors in wild-type dendrites, arguing that ligand binding is not required for receptor-mediated activation of cytoskeletal regulators. Finally, we demonstrate that ligand-receptor binding is particularly required for stabilization and maintenance of higher-order branches. Taken together, these results argue that ligand binding of receptor is dispensable for outgrowth but is required for stabilizing dendrites at appropriate locations.

## Results

### Three DMA-1 residues are critical for ligand-receptor binding

Previous work has shown that deletion of the extracellular Leucine Rich Region (LRR) domain of DMA-1, which mediates ligand binding, caused robust but disorganized dendrite branching [27]. This finding argued that the ligand complex's interaction with DMA-1 is required for dendrite patterning but is dispensable for dendrite outgrowth. To test this idea more precisely, we sought to generate point mutations in the extracellular domain of DMA-1 to abolish ligand-receptor binding. Since there is no published structure of the ligand-receptor complex, we leveraged Alphafold and surface conservation to predict the structure of the ligand-receptor complex and identify candidate residues on the receptor required for ligand binding [28]. The Alphafold2 model identified the N-terminal convex side of the DMA-1 LRR domain as the ligand-interaction surface (Fig 1c). Furthermore, surface conservation analysis showed the same region as the most conserved surface on DMA-1 (Fig 1d). Analysis of the interface in the predicted models identified that the DMA-1 residues R106, S107, and R109 were likely critical for binding to the SAX-7/L1CAM ligand (Fig 1e).

To create and validate a non-ligand interacting DMA-1 mutant *in vitro*, we established large-scale expression systems for the DMA-1 ectodomain, LECT-2/LECT2, and a single-chain version of SAX-7/L1CAM Fibronectin type III (FN) domains linked with MNR-1/FAM151B, as MNR-1/FAM151B could not be expressed alone in significant quantities. Using purified proteins and surface plasmon resonance (SPR), we showed that wild-type DMA-1 interacts strongly with a 1:1 SAX-7 FN-MNR-1:LECT-2 mixture (Fig 1f). When we engineered and tested the DMA-1 R106D-S107A-R109E mutant with SPR, we observed no ligand binding, validating that these residues are required for DMA-1 to bind the ligand complex (Fig 1f).

### DMA-1 can promote disorganized outgrowth independently of ligand binding

We used clustered regularly interspersed short palindromic repeats (CRISPR)-Cas9 genome editing to generate a non-binding receptor mutant allele [29,30]. We first asked how ligand-free receptor affects dendrite patterning and outgrowth. Given that ligand null mutants exhibit disordered dendrite outgrowth whereas receptor null mutants display little dendrite outgrowth, we predicted that PVD morphology in *dma-1* (*non-binding*) mutants would be distinct from *dma-1* null animals while resembling *sax-7/L1CAM* null animals. The *ser-2prom3::myristoylatedGFP* marker was used to label the PVD neuron, and young adult animals were imaged [26] (Fig 1a). We first compared the dendrite morphologies of *dma-1* (*non-binding*) with those of *dma-1* (*null*) and *sax-7/L1CAM* (*null*). All three mutant classes were similar in that they failed to form quaternary branches when compared to wild-type animals (Fig 2a–2d,i). We next quantified higher-order dendrite length in the proximal anterior dendrite. Interestingly, the dendrite outgrowth of *dma-1* (*non-binding*) animals phenocopied *sax-7/L1CAM* (*null*) but were distinct from *dma-1* (*null*) animals, which display severely reduced outgrowth (Fig 2b–2d,

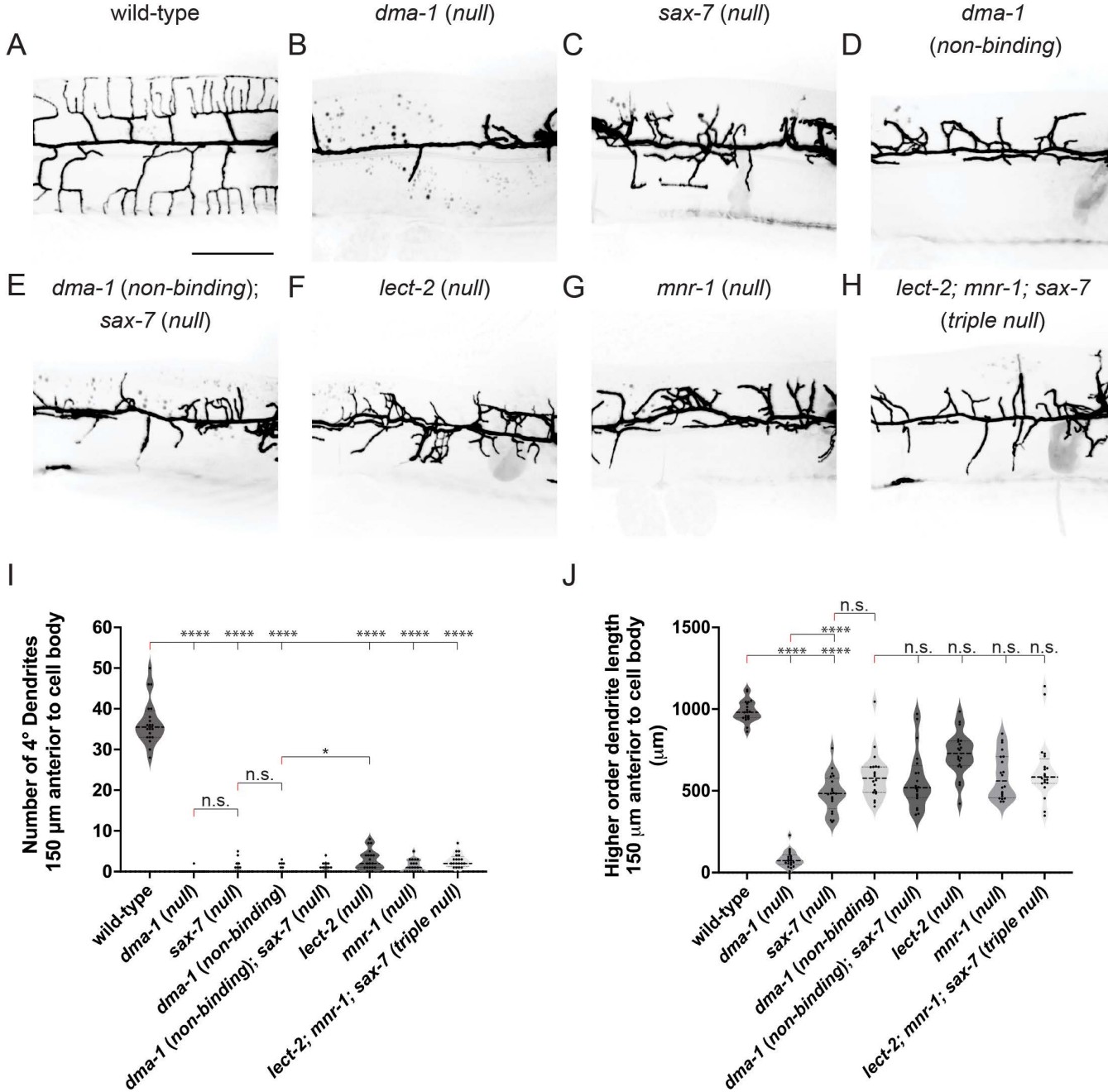

**Fig 2. Ligand-free DMA-1 receptor can promote disordered dendrite arborization.** (A-H) Fluorescence z-projections of PVD neuron 150 μm anterior to the PVD cell body in young adult animals of the genotypes indicated. PVD labelled with a *ser-2prom3::myristoylated::GFP* membrane marker. Scale bar, 50 μm. (I-J) Quantifications of number of quaternary (4°) dendrites (I) and total higher-order dendrite length (J) 150 μm anterior to the PVD cell body. Medians are represented in thick dashed lines and quartiles are represented in thin dashed lines. P values were calculated using a Brown-Forsythe and Welch one-way ANOVA with Dunnett's test. n = 20 for all conditions. n.s., (not significant), p > 0.05, *p ≤ 0.05, **p ≤ 0.01, ****p ≤ 0.0001.

j). Furthermore, *dma-1* (*non-binding*); *sax-7/L1CAM* (*null*) double mutants showed similar outgrowth phenotypes to the corresponding single mutants, arguing that our engineered point mutations indeed disrupt SAX-7/L1CAM binding (Fig 2d and 2e, j). Taken together, these data demonstrate that SAX-7/L1CAM ligand binding to DMA-1 is required for dendrite patterning but is dispensable for dendrite outgrowth.

Given that SAX-7/L1CAM is one component of the tripartite ligand complex, we next asked whether the loss of either the LECT-2/LECT2 or MNR-1/FAM151B ligands would resemble the disruption of SAX-7 binding to DMA-1 in terms of disorganized dendritic outgrowth. We compared both higher-order dendrite patterning and dendrite length in *dma-1* (*non-binding*), *lect-2/LECT2* (*null*), *mnr-1/FAM151B* (*null*), and *lect-2/LECT2; mnr-1/FAM151B; sax-7/L1CAM* triple null animals (Fig 2d and 2f–2h). All single mutant animals displayed a similar inability to form ordered quaternary branches when compared to wild-type, with *lect-2/LECT2* (*null*) showing slightly increased quaternary branches when compared to *dma-1* (*non-binding*). All mutant classes exhibited similar disordered dendrite outgrowth phenotypes in that they showed less total dendrite outgrowth than wild-type but more outgrowth than *dma-1* (*null*) (Fig 2j). To rule out the possibility that the substituted amino acids perturb intracellular trafficking of DMA-1, we introduced these mutations into an endogenously tagged *dma-1::GFP* allele and quantified fluorescence intensity in higher order dendrites [31]. We observed that non-binding receptor was present on higher order dendrites, suggesting that the mutations did not cause trafficking defects (S1a Fig). Compared to the wild-type receptor, the level of mutant receptor in the dendrites was higher (S1a and S1b Fig). Given that DMA-1 endocytosis requires the SAX-7/L1CAM ligand, we speculate that our finding can be explained by reduced endocytosis of mutant receptor due to lack of ligand binding [27]. Taken together, these data support a model in which ligand-free receptor can promote disorganized dendritic outgrowth and ligand-bound receptor is required for appropriate dendritic morphology.

To directly understand how the SAX-7/L1CAM interaction with DMA-1 instructs PVD morphology, we endogenously tagged the SAX-7/L1CAM ligand with mNeonGreen to determine the extent of dendrite misalignment with ligand in wild-type and *dma-1* (*non-binding*) animals. SAX-7/L1CAM is particularly enriched along ventral and dorsal sublateral tracts which coincide with where PVD tertiary dendrites form. Therefore, misalignment was scored by measuring the length of SAX-7/L1CAM tracts along the ventral and dorsal sublateral lines that do not colocalize with dendrites (Fig 3a and 3d). While the dendrites of wild-type animals mostly grew along the sublateral lines, leading to a low misalignment score, the misalignment in *dma-1* (*non-binding*) animals was far greater (Fig 3b, 3c and 3e). Interestingly, we observed that some secondary dendrites in wild-type animals did not overlap with SAX-7/L1CAM tracts (Fig 3c, white arrows). In contrast to secondaries that overlapped with SAX-7/L1CAM tracts, secondaries that deviated from ligand appeared to exhibit an atypical and non-perpendicular angle relative to the primary dendrite. These results strengthen our claim that the engineered point mutations disrupt SAX-7/L1CAM-DMA-1 interaction and that SAX-7/L1CAM binding to DMA-1 is not necessary to stimulate dendrite outgrowth. Furthermore, the reduced colocalization of SAX-7/L1CAM sublateral tracts with PVD dendrites in mutants supports the idea that SAX-7/L1CAM ultimately instructs the dendrite shape through its interaction with DMA-1. Finally, these data demonstrate that dendrites can grow without contacting SAX-7/L1CAM tracts, albeit in a less stereotyped manner.

## Ligand-free DMA-1 promotes outgrowth through HPO-30 and TIAM-1/TIAM1

Previous work identified that DMA-1 physically interacts with the coreceptor HPO-30, which is also required for PVD dendrite morphogenesis [18,23]. The cytosolic tail of DMA-1 and HPO-30 directly bind to the RAC guanine nucleotide exchange factor (GEF) TIAM-1/TIAM1 and the WAVE regulatory complex (WRC), respectively. Together, TIAM-1/TIAM1 and WRC promote actin polymerization by activating the Arp2/3 complex [32]. During the formation of quaternary dendrites, the DMA-1/HPO-30 complex organizes F-actin to promote dendrite outgrowth and branching [33]. Based on our work arguing that ligand-receptor binding is not required for dendrite outgrowth, we next asked whether ligand-free receptor also functioned through HPO-30 and TIAM-1/TIAM1 or if outgrowth was achieved through different interactors.

PLOS Genetics

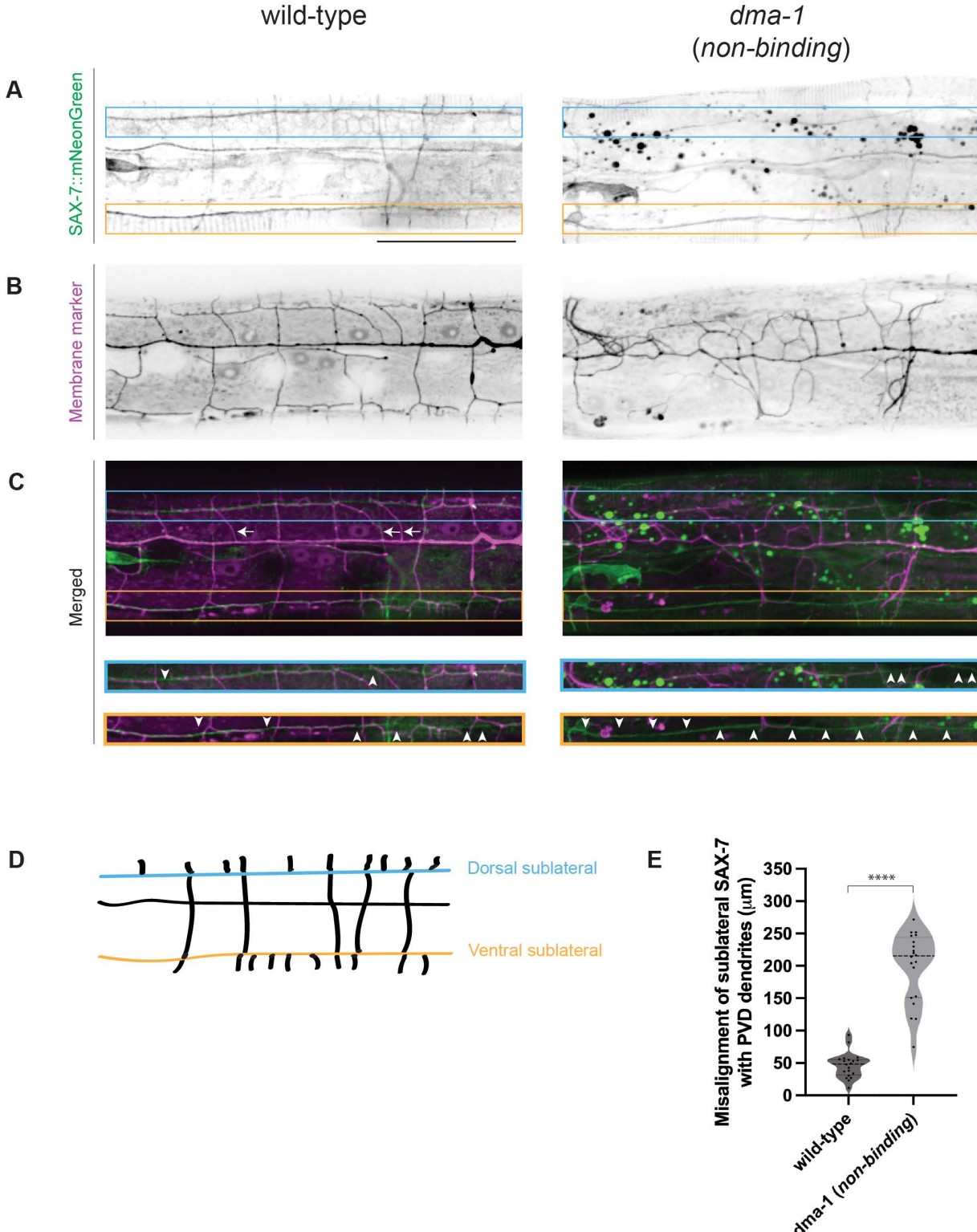

**Fig 3. Ligand-free DMA-1 receptor fails to align PVD dendrites with SAX-7/L1CAM ligand tracts.** (**A-C**) Fluorescence z-projections of endog-enously labelled SAX-7/L1CAM (**A**) and PVD morphology (**B**) 150 μm anterior to PVD cell body in young adult animals. PVD labelled with a *ser-2prom3::myristoylated::mCherry* membrane marker. Merged images shown in (**C**). Left: wild-type representative image. Right: *dma-1 (non-binding)*

representative image. Blue and orange boxes and insets denote dorsal and ventral sublateral lines, respectively. White arrows denote wild-type secondary dendrites that partially or completely fail to align with SAX-7/L1CAM tracts. White arrowheads denote regions at the sublateral lines where PVD dendrites fail to align with SAX-7 tracts. Scale bar, 50 µm. (**D**) Schematic of SAX-7/L1CAM ligand tracts 150 µm anterior to PVD cell body. Blue and orange lines represent dorsal and ventral sublateral tracts, respectively. (**E**) Quantifications measuring length of SAX-7/L1CAM ligand tract misalignment with PVD dendrites 150 µm anterior to PVD cell body. Medians are represented in thick dashed lines and quartiles are represented in thin dashed lines P value was calculated using a two-tailed unpaired *t*-test with Welch's correction. n = 20 for both conditions. ****, p ≤ 0.0001.

In a *dma-1* (*non-binding*) strain, we introduced null mutations in either *hpo-30* or *tiam-1/TIAM1* to determine if disordered dendrite outgrowth is dependent on the function of these genes. All mutant classes exhibited reduced dendrite outgrowth when compared to wild-type animals (Fig 4a and 4g). Importantly, the *dma-1* (*non-binding*); *hpo-30* (*null*) and *dma-1* (*non-binding*); *tiam-1/TIAM1* (*null*) double mutants both showed diminished dendrite outgrowth when compared to *dma-1* (*non-binding*) single mutants (Fig 4d–4g) or the *hpo-30* (*null*) and *tiam-1/TIAM1* (*null*) single mutant classes (Fig 4b, 4c and 4g). These data demonstrate that ligand-free receptor promotes dendrite outgrowth through the HPO-30 coreceptor and TIAM-1/TIAM1 Rac GEF, but our findings do not distinguish between whether these effectors function in a linear or a parallel pathway. Furthermore, our results support the surprising notion that ligand-receptor binding is not required to trigger downstream effectors that promote dendrite outgrowth and challenges the classic view that ligand-receptor signaling alone triggers neurite outgrowth.

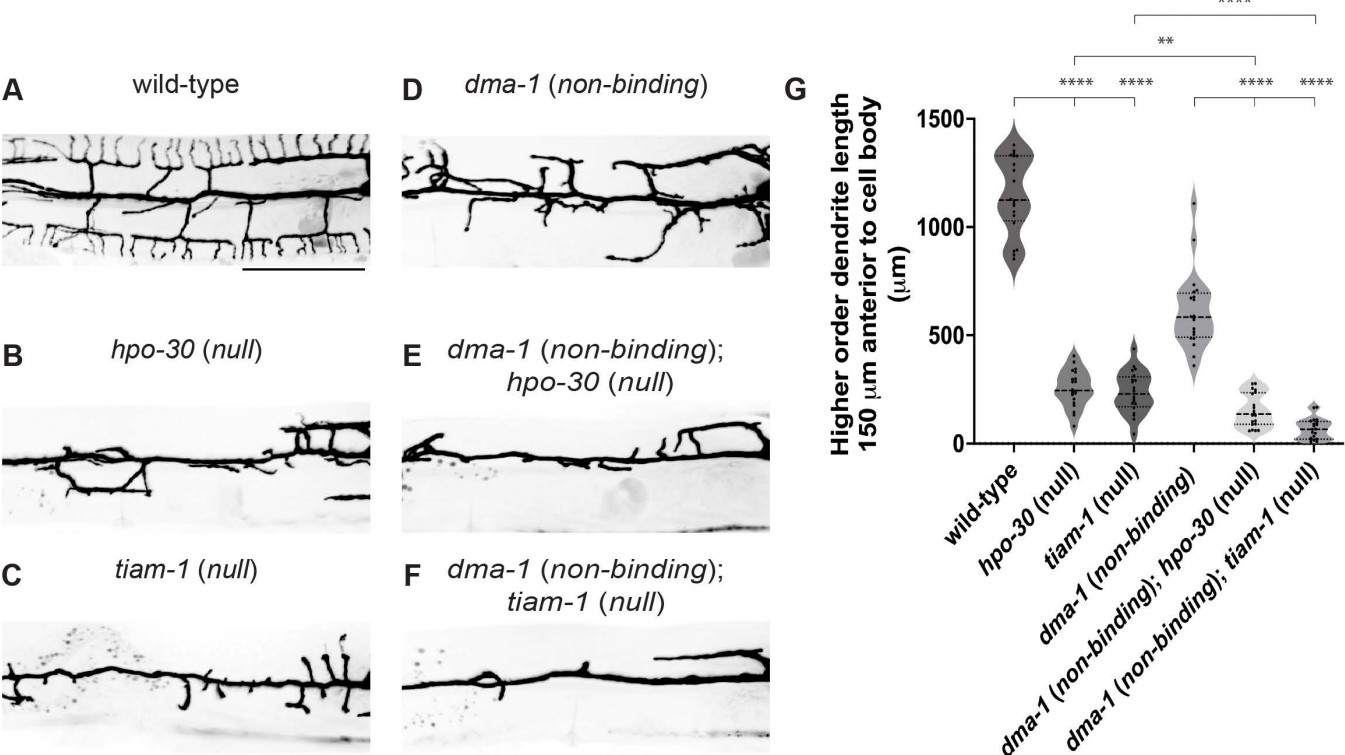

**Fig 4. Ligand-free DMA-1 receptor promotes dendrite outgrowth through the Rac GEF TIAM-1/TIAM1 and coreceptor HPO-30.** (**A-F**) Fluorescence z-projections of PVD neuron 150 µm anterior to the PVD cell body in young adult animals of the genotypes indicated. PVD labelled with a *ser-2prom3::myristoylated::GFP* membrane marker. Scale bar, 50 µm. (**G**) Quantifications of total higher-order dendrite branching length 150 µm anterior to the PVD cell body. Medians are represented in thick dashed lines and quartiles are represented in thin dashed lines. P values were calculated using a Brown-Forsythe and Welch one-way ANOVA with Dunnett's test. n = 20 for all conditions. **p ≤ 0.01, ****p ≤ 0.0001.

## Ligand binding is essential for dendrite stability and maintenance

Given that ligand-free receptor can promote dendrite outgrowth through canonical downstream effectors, we next asked how ligand-bound receptor uniquely regulates PVD morphology. Since the higher-order tertiary and quaternary branches are reduced in ligand null and *dma-1* (*non-binding*) mutant classes, we speculated that ligand-receptor binding is specifically required to maintain the stereotyped menorah shape [19,20] (Fig 2b, 2d and 2i). We formulated a model in which the ligand binding state of DMA-1 confers distinct roles: ligand-free receptors exclusively promote outgrowth, whereas ligand-bound receptors exclusively promote PVD patterning by stabilizing higher-order branches along ligand tracts. One prediction of this model is that perturbation of ligand-receptor binding will increase the dynamicity of dendritic tips due to compromised stabilization while leaving outgrowth unaffected. To test this, we performed time-lapse imaging of PVD in both wild-type and *dma-1* (*non-binding*) mutants. In support of our model, the dendritic tips of *dma-1* (*non-binding*) animals exhibited far greater growth and retraction relative to their wild-type counterparts (S2a–S2c Fig and S1 Movie).

Given that DMA-1 stabilizes dendrites during development, we further asked whether DMA-1 maintains PVD higher-order dendrites after development. To test this, we used a temperature sensitive allele of *dma-1* to temporally downregulate the receptor after menorah formation. The *dma-1* (*ts*) allele harbors a point mutation in the extracellular LRR domain (S324L). Animals cultured at the permissive temperature (16°) from hatching to adulthood display menorah formation, albeit with reduced quaternaries when compared to wild-type animals (Fig 5b and 5d). To test if DMA-1 is required for dendrite stabilization, we cultured both wild-type and *dma-1* (*ts*) animals at the permissive temperature. Upon reaching the young adult stage, when menorah formation is complete, animals were either kept at the permissive temperature or transferred to the restrictive temperature (25°). After 24 hours, morphologies of Day 1 adults were scored (Fig 5a). At permissive temperatures, we observed that *dma-1* (*ts*) animals had reduced quaternaries compared to wild-type animals, arguing that this allele is a partial loss-of-function (Fig 5d). While the shift to the restrictive temperature had no effect on dendrite morphology in wild-type animals, *dma-1* (*ts*) animals exhibited a reduced number of tertiary and quaternary branches (Fig 5b–5d). These data demonstrate that DMA-1 is not only required for PVD development but also for dendrite branch maintenance.

We hypothesized that only ligand-bound DMA-1 receptor could promote branch stabilization. To test this model, we generated a *dma-1* allele where an auxin inducible degron (AID*) sequence was inserted into an endogenously tagged *dma-1::GFP* allele [31]. In tandem with a pan-somatic transgene expressing the F-box protein TIR1, our degron tagged *dma-1::GFP::AID** allele conferred the ability to temporally degrade receptor upon addition of the plant hormone auxin [34]. We determined that 17 hours of auxin treatment was sufficient to induce significant degradation of DMA-1 based on loss of the endogenous GFP signal (S3a and S3b Fig). To assess if the induced degradation affected DMA-1 function, we treated young adult *dma-1::GFP::AID** animals with 10 mM auxin and observed a loss of quaternary branches (Fig 6b' and 6c). We next tested whether ligand-free receptor could promote higher-order branch stabilization. Given that *dma-1* (*non-binding*) homozygous animals are unable to form higher-order branches, we generated transheterozygotes that had one copy of a non-degradable *dma-1* (*non-binding*) allele and one copy of the *dma-1::GFP::AID** allele (Figs 2i and 6a). These transheterozygotes provided a system in which higher-order branches could form by the young adult stage, allowing us to test the role of ligand binding in dendrite maintenance (Fig 6b'''). As a control, we generated a second class of transheterozygotes containing one copy of a wild-type, non-degradable *dma-1* allele and one copy of the *dma-1::GFP::AID** allele (Fig 6a). We predicted that only the transheterozygotes with a non-degradable wild-type allele would maintain dendrite branches after the depletion of degradable receptor. Temporal downregulation of the AID* tagged pool of receptor, which can bind ligand, was induced by transferring young adult transheterozygotes from auxin free plates to 10 mM auxin plates, and Day 1 adults were imaged and scored for quaternary branches (Fig 6a). With this experimental paradigm, the majority of remaining receptor in the transheterozygotes would either be wild-type or *dma-1* (*non-binding*) after degradation. Transheterozygotes expressing a non-degradable wild-type receptor allele exhibited no loss of quaternary branching after auxin treatment (Fig 6b'' and 6c). In contrast, transheterozygotes expressing the non-binding receptor allele showed reduced quaternary

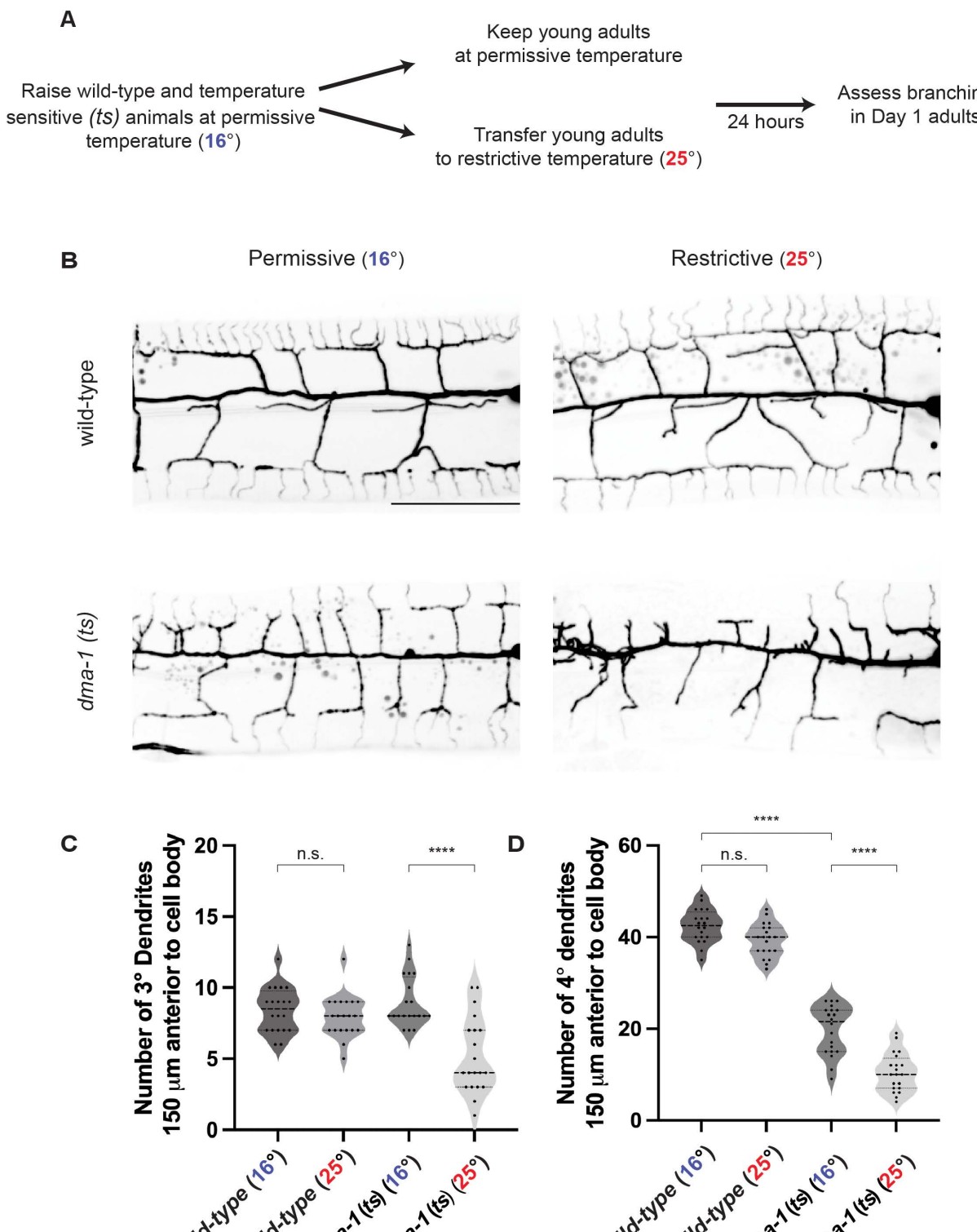

**Fig 5. The DMA-1 receptor is required for higher-order dendrite branch maintenance.** (**A**) Experimental schematic to determine how temporal downregulation of DMA-1 receptor after PVD outgrowth affects higher-order dendrite branching. (**B**) Fluorescence z-projections of PVD neuron 150 μm anterior to the PVD cell body in Day 1 adult animals. PVD labelled with a *ser-2prom3::myristoylated::GFP* membrane marker. Top, representative

wild-type animals. Bottom, representative *dma-1* temperature sensitive (*ts*) animals. Left, animals cultured at permissive temperature (16°). Right, animals transferred to restrictive temperature (25°) at the young adult stage. Scale bar, 50 μm. (**C-D**) Quantifications of number of tertiary (3°) dendrites (**C**) and quaternary (4°) dendrites (**D**) 150 μm anterior to the PVD cell body. Medians are represented in thick dashed lines and quartiles are represented in thin dashed lines. P values were calculated using a Brown-Forsythe and Welch one-way ANOVA with Dunnett's test. n = 20 for all conditions. n.s., (not significant), p > 0.05, ****p ≤ 0.0001.

branches after auxin treatment (Fig 6b''' and 6c). Taken together, these data argue that ligand-receptor binding is required for the stabilization and maintenance of higher-order dendrites.

## Discussion

The acquisition of appropriate dendritic morphology is critical for a neuron's ability to receive information. Underlying this process is the coordination between growth and stabilization of dendritic branches during development. Critical questions in developmental neurobiology include how dendrite growth, branching, and stabilization generate stereotyped dendrite morphology and how these events are regulated by guidance molecules and their cognate receptors. One possibility is that molecular guidance cues promote a stereotyped outgrowth program, in which dendrite growth only occurs at locations where ligands are enriched and can tether to their cognate receptors. Alternatively, dendrites may employ a selective stabilization program in which processes exhibit stochastic growth, branching, and retraction events, but only stabilize at regions with high levels of ligand. The stereotyped growth model predicts that guidance receptor activation leads to directed outgrowth, and that the loss of guidance cues severely inhibits dendrite outgrowth. In contrast, the selective stabilization model predicts that only a fraction of the dendrite outgrowth events leads to a stabilized branch, and that the loss of guidance cues will prevent stabilization of branches at target locations while having either a small or no effect on dendrite outgrowth [35,36]. Here, we propose that the PVD neuron employs a selective stabilization program in the formation of its higher order dendrites. Our findings reveal that the DMA-1 receptor can regulate both outgrowth and stabilization: in the ligand-free state, DMA-1 promotes stochastic outgrowth; in the ligand-bound state, DMA-1 promotes stereotyped dendritic morphology by stabilizing dendrites along ligand tracts, which enables further dendritic outgrowth at these regions (Fig 7).

Our claim that a receptor can promote dendritic outgrowth without binding ligand is supported by several lines of evidence. Although ligand null mutants lack the menorah structure, they can establish an extensive disorganized dendritic arbor that likely arises from stochastic dendritic outgrowth (Fig 2i and 2j). Furthermore, mutants in which the extracellular LRR domain of DMA-1 was deleted also exhibited disorganized branching [27]. In this work, we demonstrate that the substitution of three residues in the LRR domain is sufficient to disrupt ligand binding, and that the disorganized dendritic morphologies by mutants expressing this non-binding receptor allele phenocopies those found in ligand null mutants (Figs 1e, 1f and 2d). Examining ligand localization lends further support for the claim that ligand-free receptors can promote dendrite outgrowth. The SAX-7/L1CAM ligand is distributed unevenly along the epidermal cells upon which PVD elaborates its dendrites: SAX-7/L1CAM is relatively low in regions where secondary dendrites extend but is high in regions where tertiary and quaternary dendrites develop [20]. These findings are inconsistent with the model that dendrite outgrowth only occurs along ligand tracts. Instead, the SAX-7/L1CAM distribution pattern is consistent with the growth, retraction, and stabilization dynamics of secondary dendrite formation [20]. We posit that the subgroup of secondaries which traverse through ligand deficient regions employ ligand-free DMA-1 receptor to drive stochastic outgrowth. Within this subgroup, secondaries that reach ligand at sublateral tracts employ ligand-bound DMA-1 to stabilize. In contrast, secondaries that fail to reach the sublateral tracts are retracted.

Tertiary and quaternary dendrites extend in ligand rich regions, which presents the following challenge: how do these processes avoid stabilizing to an extent that compromises outgrowth? We propose that these branches employ both ligand-bound and ligand-free receptor to balance stabilization and outgrowth. This theory is supported by experimental

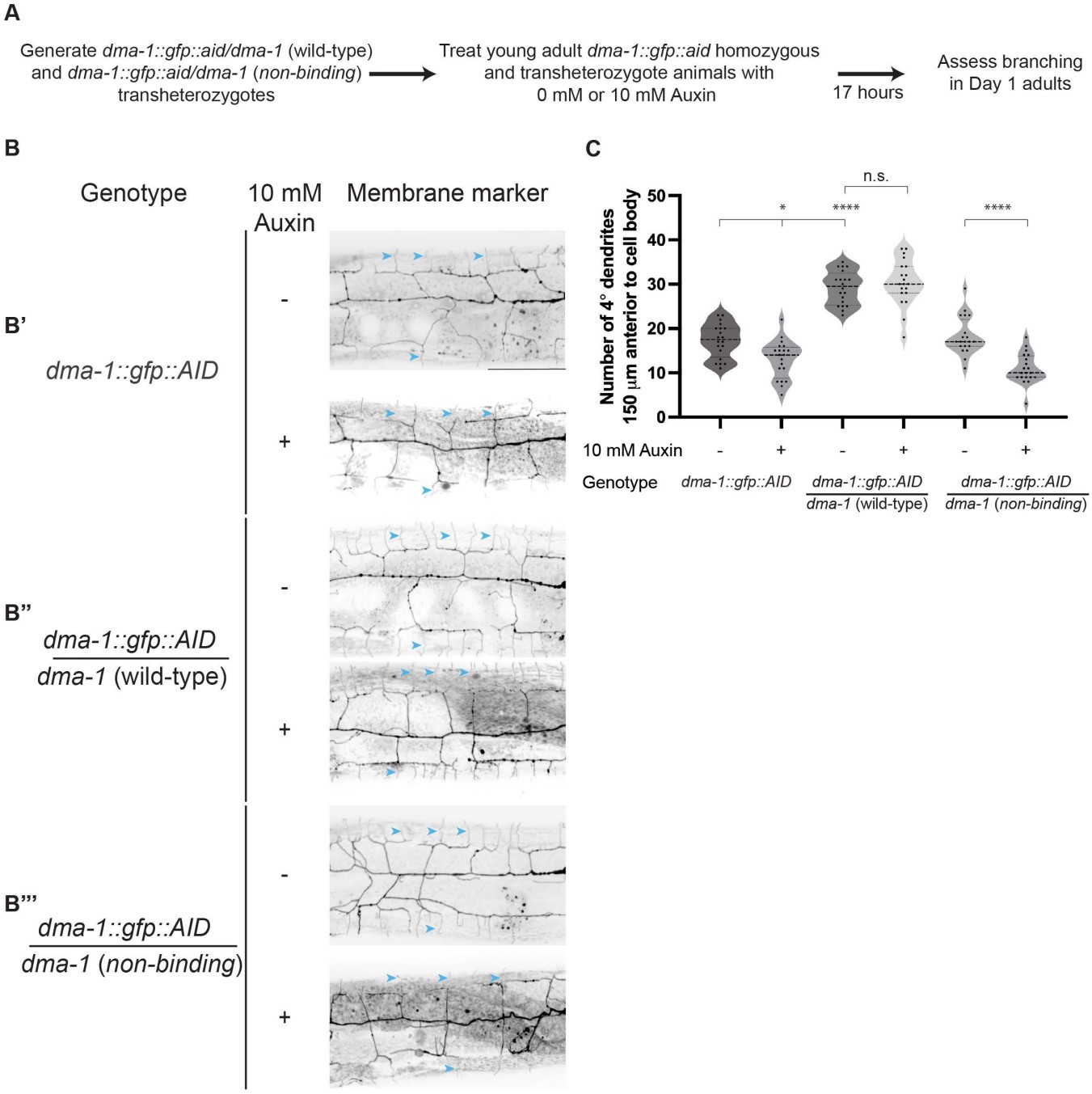

**Fig 6. The DMA-1 receptor promotes higher-order branch stabilization through ligand binding.** (A) Experimental schematic to determine whether ligand-free receptor promotes branch stabilization. (B) Fluorescence maximum intensity z-projections of PVD in Day 1 adult animals. PVD labelled with a *ser-2prom3::myristoylated::mCherry* membrane marker. B', *dma-1::gfp::aid* homozygous animals. B," *dma-1::gfp::aid/dma-1* (wild-type) transheterozygotes. B"', *dma-1::gfp::aid/dma-1* (*non-binding*) transheterozygotes. Top images in B', B," and B"' are representative images of animals that did not receive auxin treatment. Bottom images are representative images of animals that received 10 mM auxin treatment at the young adult stage. Blue arrowheads denote examples of quaternary branches. Scale bar, 50 μm. (C) Quantifications of quaternary (4°) branches 150 μm anterior to the PVD cell body. Medians are represented in thick dashed lines and quartiles are represented in thin dashed lines. P values were calculated using a Brown-Forsythe and Welch one-way ANOVA with Dunnett's test. n = 20 for all conditions. n.s., (not significant), p > 0.05, *p ≤ 0.05,**** p ≤ 0.0001.

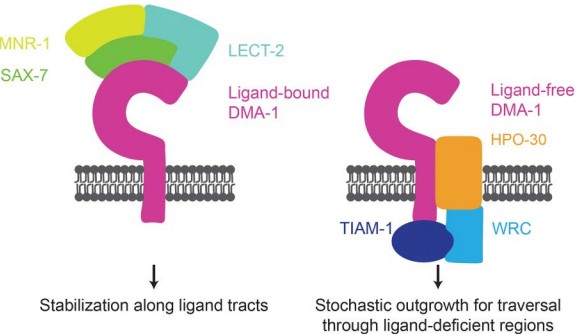

**Fig 7. Model for how ligand-bound and ligand-free receptor pools promote PVD dendrite arborization.** DMA-1 receptor binding to the SAX-7/LECT-2/MNR-1 tripartite ligand complex promotes dendrite branch stabilization at ligand tracts. Ligand-free receptor drives stochastic dendrite outgrowth branching through ligand deficient regions and functions through the downstream effectors TIAM-1/TIAM1 and HPO-30.

evidence: previous work from our group demonstrates that the DMA-1 receptor is endocytosed in a manner dependent on ligand binding [27]. It is unlikely that the transmembrane ligand complex enters the endosome, arguing that the endocytosed receptor is in the ligand-free state. We speculate that this ligand-free receptor pool is recycled back to the plasma membrane to drive outgrowth. Indeed, inhibition of receptor endocytosis yields severely truncated dendrite arbors, which we predict is caused by a diminished ligand-free receptor pool capable of driving outgrowth [27]. Therefore, we propose that a cyclical process consisting of ligand-receptor binding, receptor endocytosis, and recycling of ligand-free receptor to the plasma membrane drives tertiary and quaternary branch growth and stabilization. This model can also explain the genetic interaction between the partial loss of *kpc-1* and partial loss of *mnr-1*: the truncated tertiary dendrite morphology of *kpc-1* mutants can be partially rescued by lowering MNR-1 dosage [37]. In *kpc-1* mutants, plasma membrane localized DMA-1 increases whereas endosomal DMA-1 decreases [27]. It is possible that when KPC-1 activity is attenuated, the lowering of functional MNR-1 both reduces the adhesion of plasma membrane localized DMA-1 and promotes receptor endocytosis. This rebalancing of the two pools of DMA-1 enables the partial rescue of the truncated tertiary dendrites.

While we demonstrate that the DMA-1 receptor can function in a ligand-independent manner to regulate dendrite morphogenesis, other systems exhibit a clear requirement for ligand-receptor binding. For example, mutants defective for either the Robo receptor or Slit ligand exhibit similar defects in dendrite arborization of the *Drosophila* aCC motoneuron [38]. This finding suggests that the Slit ligand might promote dendrite arborization by activating the Robo receptor. Studies on the *Drosophila* class IV dendritic arborization (C4da) neurons show a similar dependency on ligand-receptor binding: both the Frazzled receptor and Netrin ligand are required for local dendrite growth towards the source of ligand [39]. Moreover, the secreted TGF-β ligand Maverick and its cognate receptor tyrosine kinase Ret both promote C4da dendrite arborization [40]. In these developmental contexts, ligand binding is required to activate an otherwise inert receptor.

There are examples of receptors functioning independently of ligand binding. Genetic evidence demonstrates that the SAX-3/Robo receptor regulates axon guidance of the *C. elegans* AWB neuron independently of its cognate ligand SLT-1/Slit [41]. Furthermore, studies in a polarized epithelial cell demonstrate that the UNC-40/DCC receptor can form clusters and recruit downstream effectors to drive F-actin formation without binding its cognate ligand UNC-6/netrin [42]. These cases demonstrate that the activation of some receptors is not contingent on binding to a cognate ligand.

Taken together, our findings support a model in which dendrite arborization can be achieved through both a cell-intrinsic program which promotes stochastic growth and branching and a selective stabilization mechanism. We propose that ligand-free receptor promotes stochastic outgrowth by promoting traversal through ligand-deficient areas, and that ligand-bound receptor stabilizes dendrites along ligand tracts to subsequently enable further dendritic outgrowth at

appropriate regions. The balancing of these programs enables the PVD neuron to navigate a complex tissue environment and ultimately innervate appropriate regions to form a complex dendritic arbor.

## Materials and methods

### Protein structure modeling

We used Alphafold 2 (version 2.3) locally, as implemented in Colabfold version 1.5.2 [28,43,44]. Alphafold 2 reported an iPTM value of 0.806 for the predicted complex of DMA-1 with SAX-7/L1CAM, while other binary complexes of DMA-1 were not predicted to have similarly high iPTM values. While Alphafold 3 runs predicted a significantly different model, a similar set of DMA-1 residues (including R106 and R109) were involved in SAX-7/L1CAM binding, in agreement with the importance of this conserved DMA-1 surface for ligand binding. For surface conservation analysis, we used the Consurf server, available at https://consurf.tau.ac.il [45].

### Protein expression and purification

We used the baculoviral expression system for protein production *in vitro*. In short, we created baculoviruses using co-transfection of Sf9 cells with linearized baculoviral DNA with plasmids containing the protein of interest in the pAcGP67A backbone, which carries a gp64 signal peptide for protein secretion. The amplified viral stocks were used to infect High Five cells (*Trichoplusia ni*) grown in serum-free ESF-921 (Expression Systems). The secreted proteins, in which LECT-2 was N-terminally hexahistidine tagged and the remaining proteins were C-terminally hexahistidine tagged, were purified using Ni-NTA metal affinity chromatography, followed by size-exclusion chromatography on Superdex 200 Increase 10/300 columns in HEPES-buffered saline (HBS), containing 10 mM HEPES pH 7.4 and 150 mM NaCl.

For SPR experiments we expressed and purified the following four constructs: wild-type and mutated DMA-1 ecto-domain (residues 20–505) with a C-terminal Avi-tag and hexahistidine tag; LECT-2/LECT2 (residues 23–335) with an N-terminal hexahistidine tag followed by a tobacco etch virus protease site; and a single-chain fusion of SAX-7/L1CAM Fibronectin type III domains (SAX-7 FN) (residues 633–1207) followed by a 34-residue $(GGGS)_8GG$ linker, MNR-1/FAM151B with the GPI-anchor sequence removed (residues 17–457) and a C-terminal hexahistidine tag. All constructs carried an N-terminal gp64 signal peptide for secretion.

### Surface plasmon resonance

For SPR, we used a Biacore T200 machine and a streptavidin-coupled Series S Sensor Chip SA (Cytiva). Wild-type and mutant DMA-1 ectodomains were biotinylated at the biotin acceptor peptide (i.e., Avi-tag) using BirA biotin ligase. Approximately 200 response units of wild-type and mutant DMA-1 were captured on separate channels, and 1:1 mixed SAX-7 FN–MNR-1:LECT-2 samples were run over both channels in a buffer containing 10 mM HEPES, pH 7.4, 150 mM NaCl and 0.05% Tween-20.

### *C. elegans* culture and strains

*C. elegans* were cultured at 20°C on NGM plates using OP50 *Escherichia coli* as a food source according to standard procedures unless otherwise noted [46]. Two constructs were used as PVD membrane markers: *ser-2prom3::myristoylated::GFP* (*wyIs*592) and *ser-2prom3::myristoylated::mCherry* (*wyIs*581). All strains and primers used in this study are listed in Table 1.

### Generation of genome-edited strains

Point mutations and endogenous fluorophore or degron insertions were created by gonadal microinjection of CRISPR-Cas9 protein complexes. For the generation of the *dma-1* (*non-binding*) allele, a single-stranded DNA primer consisting of 30

**Table 1. Key resources table.**

| Reagent type (species) or resource | Designation | Source or reference | Identifiers | Additional information |
|---|---|---|---|---|
| Strain, strain background (*C. elegans*) | wyIs592 [ser-2prom3:: myristoylated::GFP] III | [51] | TV15911 | Referred to as *wild-type* in Figs 1, 2, 4 and 5. |
| Strain, strain background (*C. elegans*) | dma-1 (wy686) I; wyIs592 III | [26] | TV18862 | Referred to as *dma-1 (null)* in Fig 2 |
| Strain, strain background (*C. elegans*) | wyIs592 III; sax-7 (nj48) IV | [20] | TV17248 | Referred to as *sax-7 (null)* in Fig 2 |
| Strain, strain background (*C. elegans*) | dma-1 (wy1907) I; wyIs592 III | This study | TV29063 | Referred to as *dma-1 (non-binding)* in Figs 2 and 4 |
| Strain, strain background (*C. elegans*) | dma-1 (wy1907) I; wyIs592 III; sax-7 (nj48) IV | This study | TV29144 | Referred to as *dma-1 (non-binding); sax-7 (null)* in Fig 2 |
| Strain, strain background (*C. elegans*) | lect-2 (wy953) II; wyIs592 III | [51] | TV18540 | Referred to as *lect-2 (null)* in Fig 2 |
| Strain, strain background (*C. elegans*) | wyIs592 III; mnr-1 (wy758) V | [20] | TV16271 | Referred to as *mnr-1 (null)* in Fig 2 |
| Strain, strain background (*C. elegans*) | lect-2 (wy953) II; wyIs592 III; sax-7 (nj48) IV; mnr-1 (wy758) V | This study | TV23848 | Referred to as *lect-2; mnr-1; sax-7 (triple null)* in Fig 2 |
| Strain, strain background (*C. elegans*) | sax-7 (wy1982), wyIs581 [ser-2prom3::myristoylated:: mCherry] IV | This study | TV29421 | wy1982 = SAX-7 ::mNeonGreen::AID* Referred to as *wild-type* in Fig 3 |
| Strain, strain background (*C. elegans*) | dma-1 (wy1907) I; sax-7 (nj48), wyIs581 IV | This study | TV29684 | Referred to as *dma-1 (non-binding)* in Fig 3 |
| Strain, strain background (*C. elegans*) | wyIs592 III; hpo-30 (ok2047) V | [18] | TV16821 | Referred to as *hpo-30 (null)* in Fig 4 |
| Strain, strain background (*C. elegans*) | tiam-1 (tm1556) I; wyIs592 III; wyEx4288 | [18] | TV17689 | wyEx4288 = dma-1 fosmid overexpression. Animals were selected for imaging based on the absence of coinjection marker (myo-2 > mCherry). Referred to as *tiam-1 (null)* in Fig 4 |
| Strain, strain background (*C. elegans*) | dma-1 (wy1907) I; wyIs592 III; hpo-30 (ok2047) V | This study | TV29164 | Referred to as *dma-1 (non-binding); hpo-30 (null)* in Fig 4 |
| Strain, strain background (*C. elegans*) | dma-1 (wy1953), tiam-1 (tm1556) I; wyIs592 III | This study | TV29232 | Referred to as *dma-1 (non-binding); tiam-1 (null)* in Fig 4 |
| Strain, strain background (*C. elegans*) | dma-1 (wy50544) I; wyIs592 III | This study | TV25787 | wy50544 = S324L. Referred to as *dma-1 (temperature sensitive)* in Fig 5 |
| Strain, strain background (*C. elegans*) | wrdSi22 [eft-3p::TIR1::F2A: BFP::tbb-2 3'UTR + SEC] I; wyIs581 IV | This study | TV29427 | |
| Strain, strain background (*C. elegans*) | dma-1 (wy1907), wrdSi22 I; wyIs581 IV | This study | TV29705 | |
| Strain, strain background (*C. elegans*) | dma-1 (wy2039), wrdSi22 I; wyIs581 IV | This study | TV29853 | Referred to as *dma-1::gfp::aid* in Fig 6. Crossed males of this strain with hermaphrodites of either TV29427 or TV29705 to generate the two classes of transheterozygotes |
| Strain, strain background (*C. elegans*) | dma-1 (wy1246) I; wyIs581 IV | [31] | TV244661 | Referred to as wild-type in S1 Fig |
| Strain, strain background (*C. elegans*) | dma-1 (wy1932) I; wyIs581 IV | This study | TV29120 | Referred to as *dma-1 (non-binding)::gfp* in S1 Fig |
| Sequence based reagent | dma-1 point mutation sgRNA | This study | | AGATGGCAGTCGAATGGATC. Used to generate alleles wy1907 and wy1953 |
| Sequence based reagent | sax-7 knock-in sgRNA | This study | | CCGGCCGAACGGCCCGAGA A. Used to generate the wy1982 allele |
| Sequence based reagent | dma-1 knock-in sgRNA | This study | | ATGAACTATACAAAGGTACC. Used to generate the wy2039 allele |

*(Continued)*

**Table 1.** (Continued)

| Reagent type (species) or resource | Designation | Source or reference | Identifiers | Additional information |
|---|---|---|---|---|
| Sequence based reagent | dma-1 R106D, S107A, R109E repair primer | This study | | AGTTCTTCGTCTCATCAAT TGTCAGATTCCGGCCATGT CAGACGCCATTGAACTGCCA TCTTTGGAAGTATTAGATCT ACAC. Includes 30 nucleotide homology arms. Repair primer introduces the R106D, S107A, and R109 E mutations used to generate alleles wy1907 and wy1953. Primer also introduces a silent site mutation conferring an HaeIII restriction site for genotyping. |
| Sequence based reagent | Primer dma-1 (R106D, S107A, R109E) genotyping forward | This study | | TCTATTGCACTACCATC ATCATGTCCG |
| Sequence based reagent | Primer dma-1 (R106D, S107A, R109E) genotyping reverse | This study | | AATGGCATCGTTCGGAACGC. Amplifies a 954 nucleotide band in tandem with Primer dma-1 (R106D, S107A, R109E) genotyping forward. To genotype wy1907 and wy1953, digest with HaeIII to obtain 621 and 333 nucleotide bands. |
| Sequence based reagent | Primer sax-7 (nj48) genotyping forward | This study | | GAAATACACACAAATACGA GTGCTG |
| Sequence based reagent | Primer sax-7 (nj48) genotyping reverse | This study | | TGTGTACTTTTTGTTGGCAAAC AAAAATACAC. Amplifies an 813 nucleotide band for the wild-type allele and 245 nucleotide band for the (nj48) allele in tandem with Primer sax-7 (nj48) genotyping forward. |
| Sequence based reagent | Primer sax-7 (wy1982) knock-in forward | This study | | CTGTTTCAGGTCAATACGTTC CACAAAAGA GCTTGATGCCA GCTGAGCGACCAGAAAAAGG ATCAACGTCGACGTTTGTCGG TGGCGGTGGATCGGGAGG |
| Sequence based reagent | Primer sax-7 (wy1982) knock-in reverse | This study | | TGAAAGTACACATAAATAAATA AGCTCAAGCAATTTCGTGTAA TAAAATGAAAAAGAAGGAACA CGTGGATAGTCTTCTACTTC ACGAACGCCGCCGCCT. Amplifies a repair template from pJW2171 in tandem with Primer sax-7 (wy1982) knock-in forward. |
| Sequence based reagent | Primer hpo-30 (ok2047) genotyping forward | This study | | TGTTTGGACAGTAAAATCTAATT TATTGTTACCGC |
| Sequence based reagent | Primer hpo-30 (ok2047) genotyping reverse | This study | | GCTACGCTTGTAATGTACTC CTATGGT |
| Sequence based reagent | Primer dma-1 (wy2039) knock-in forward | This study | | CTGGGATTACACATGGCATG GATGAACTATACAAAGGATCC GGAGGTGGCGGGATGCCTA AAGATCCAGC |
| Sequence based reagent | Primer dma-1 (wy2039) knock-in reverse | This study | | AGAGTGTTCTCGTGTCACATAT GATCCTTTACTTCCGCTGCCA CTACCGGTACCCTTCACGAAC GCCGCCGCCTCCGGGCCAC CGCTTG. Amplifies a repair template from pJW2098 in tandem with Primer dma-1 (wy2039) knock-in forward. |
| Sequence based reagent | Primer dma-1 (wy2039) genotyping forward | This study | | CTGCCCTTTCGAAAGATCCC AAC |
| Sequence based reagent | Primer dma-1 (wy2039) genotyping reverse | This study | | TAGACAGGAGCCTTAATCAG CAGC. Amplifies a 699 nucleotide band in tandem with Primer dma-1 (wy2039) genotyping forward. |

*(Continued)*

**Table 1.** (Continued)

| Reagent type (species) or resource | Designation | Source or reference | Identifiers | Additional information |
|---|---|---|---|---|
| Recombinant DNA reagent | Plasmid: pJW2171 | [34] | | Vector for cloning homologous repair template |
| Recombinant DNA reagent | Plasmid: pJW2098 | [34] | | Vector for cloning homologous repair template |

nucleotide homology arms was used as a repair template. This repair template included a silent site mutation conferring an *HaeIII* restriction enzyme cut site and removed the PAM site to prevent Cas9 from editing the repair template. SAX-7 was C-terminally tagged with mNeonGreen::AID*. The repair template for this insertion was generated by PCR amplification from the pJW2171 plasmid and contained a GSGGGG linker between the mNeonGreen and AID* sequences [34]. Generation of the *dma-1::GFP::AID** allele was achieved by inserting an AID* sequence to the C terminus of the GFP sequence, which itself was previously inserted into the juxtamembrane region of DMA-1 [31]. The repair template for this insertion was generated by PCR amplification from the pJW2098 plasmid and included a GSGGGG linker upstream of the AID* sequence [34].

CRISPR-Cas9 genome editing was performed using standard protocols [47]. Cas9 protein and tracRNA (IDT) were both injected at 1.525 µM concentrations. sgRNAs (IDT) were injected at a concentration of 1.525 µM. Repair templates were injected at a concentration of 5 µM. To select for candidate animals with a successful genome edit, the pRF4 plasmid was injected at 50 ng µL$^{-1}$. $F_1$ roller animals were singled to fresh plates, and $F_2$ animals were screened for the desired genome edit using PCR and Sanger sequencing. Primers used to amplify homology repair templates, sgRNA sequences, and genotyping primers used to verify generation of desired edits are listed in Table 1.

## *C. elegans* confocal microscopy and image analysis

Young and Day 1 adult hermaphrodite *C. elegans* animals were anaesthetized using 10 mM levamisole (Sigma-Aldrich) in M9 buffer and mounted on 3% agarose pads. For time-lapse microscopy, worms were anaesthetized using 10 mM levamisole and mounted on 5% agarose pads before imaging on a 35-mm glass-bottom dish (MatTek Corp). With the exception of animals heat shocked at a 25°C incubator, all imaged animals were maintained in a 20°C incubator.

*C. elegans* animals were imaged on imaging systems consisting of either (1) an inverted Zeiss Axio Observer Z1 microscope paired with a Yokogawa CSU-X1 spinning-disk unit, a Hamamatsu EM-CCD digital camera, 488 nm and 561 nm solid state lasers, and a C-Apochromat 40x 0.9 NA objective controlled by Metamorph (version 7.8.12.0) or (2) an inverted Zeiss Axio Observer Z1 microscope paired with a Yokogawa CSU-W1 spinning-disk unit, a Prime 95B Scientific CMOS camera, 488 nm and 561 nm solid state lasers, and a C-Apochromat 40x 0.9NA objective controlled by 3i Slidebook (v6). The length of z-stacks was determined by manually setting top and bottom slices. For time-lapse microscopy, images were acquired at 2-minute intervals for 20 minutes. These images were quantified by scoring 20 dendritic tips using 5 animals each for a total of 100 analyzed dendrites. In all cases, imaging conditions such as laser power, exposure time, and step-size (0.33 µm) were identical for all genotypes and conditions across the experiment.

Image analysis was performed on unprocessed images using Fiji software [48]. Maximum-intensity and sum-intensity projections were rotated, cropped, and straightened to generate display images, and brightness and contrast were adjusted in Fiji. PVD micrographs of individual animals were stitched and cropped to show the region 150 µm anterior to the cell body using Fiji [49]. All images are oriented in the following manner: the anterior-posterior axis is left to right, and ventral-dorsal axis is bottom to top.

The Simple Neurite Tracer (SNT) plugin was used in Fiji to calculate higher-order dendrite length [50]. Briefly, non-overlapping traces were drawn on PVD higher-order dendrites for individual animals and saved. SNT was used to measure the length of individual traces.

## Statistical analysis

Statistical analysis was performed in GraphPad Prism 10. Animals were selected for measurements based on developmental stage, orientation on slide, and health. Sample size refers to the number of *C. elegans* animals imaged. For statistical tests, single pairwise comparisons of genotypes or treatments were analyzed using either two-tailed unpaired Student's *t*-tests or two-tailed unpaired *t*-test with Welch's correction. Multiple comparisons were performed using one-way analysis of variance (ANOVA) followed by post hoc Dunnett's test. Figure legends indicate sample size, statistical tests used, and *p* values. All graphs prepared in GraphPad Prism.

## Supporting information

**S1 Fig. Non-binding DMA-1 receptor robustly localizes to higher order dendrites.** (**A**) (Top) Fluorescence sum intensity z-projections of endogenously labelled wild-type (left) or non-binding (right) DMA-1 receptor in Day 1 Adults. (Bottom) Fluorescence maximum intensity z-projections of PVD. Red arrowheads indicate examples of gut granules which exhibit autofluorescence and are not present in the PVD. Scale bar, 50 µm. (**B**) Quantification of receptor fluorescence intensity in higher order dendrites. Medians are represented in thick dashed lines and quartiles are represented in thin dashed lines. P value was calculated using a two-tailed unpaired Student's t-test. n = 20 for all conditions. **** p < 0.0001.
(TIF)

**S2 Fig. Ligand-receptor binding restricts excess growth and retraction of dendrites.** (**A**) Time series montage showing six frames of fluorescence maximum intensity z-projections of PVD at 4-min intervals in wild-type (left) and *dma-1 (non-binding)* (right) L4 animals. Growth and retraction events relative to the previous time frame are indicated with green and magenta arrowheads, respectively. Still images are from S1 Movie and represent cropped dorsal higher order dendrites. Scale bar, 10 µm. (**B-C**) Quantifications of total growth (**B**) and retraction (**C**) of individual dendritic tips over 20 minutes. Medians are represented in thick lines and quartiles are represented in thin dashed lines. P values were calculated using a two-tailed unpaired Student's t-test. n = 100 for both conditions. **** p < 0.0001.
(TIF)

**S3 Fig. Temporal degradation of degron tagged DMA-1.** (**A**) Fluorescence sum intensity z-projections of endogenously labelled DMA-1::GFP::AID treated without auxin (top) or with 10 mM Auxin (bottom). Red arrowheads indicate examples of gut granules which exhibit autofluorescence and are not present in the PVD neuron. Scale bar, 50 µm. (**B**) Quantifications of DMA-1::GFP::AID fluorescence intensity in the PVD cell body. Medians are represented in thick dashed lines and quartiles are represented in thin dashed lines. P value was calculated using a two-tailed unpaired Student's *t*-test. n = 20 for all conditions. *p ≤ 0.05.
(TIF)

**S1 Movie. Visualizing dendritic tip dynamicity in wild-type and *dma-1 (non-binding)* animals.** Time-lapse imaging of PVD dendrites 150 µm anterior to the cell body in wild-type (top) or *dma-1 (non-binding)* (bottom) L4 animals. Green and magenta carets denote areas of growth and retraction, respectively. Scale bar, 50 µm.
(AVI)

## Acknowledgments

We would like to thank the past and current members of the Shen lab who provided excellent input in the writing of this manuscript. In particular, we thank Callista Yee, Kelsie Eichel, Yue Sun, Dane Kawano, and Junhao Xu for their feedback. We thank the trainees, staff, and faculty of the Stanford Department of Biology who provided helpful scientific discussions, technical assistance, and support. We also would like to thank the Stanford Superworm community for their encouragement and feedback. Finally, we thank Bryan Kirsch for critical reading of the manuscript.

## Author contributions

**Conceptualization:** Anay R. Reddy, Kang Shen.

**Data curation:** Anay R. Reddy, Sebastian J. Machera.

**Formal analysis:** Anay R. Reddy, Sebastian J. Machera.

**Investigation:** Anay R. Reddy, Sebastian J. Machera, Zoe T. Cook, Huichao Deng, Wioletta I. Nawrocka.

**Methodology:** Anay R. Reddy, Sebastian J. Machera, Zoe T. Cook, Huichao Deng, Wioletta I. Nawrocka.

**Project administration:** Engin Özkan, Kang Shen.

**Resources:** Anay R. Reddy, Sebastian J. Machera, Zoe T. Cook, Huichao Deng.

**Supervision:** Engin Özkan, Kang Shen.

**Visualization:** Anay R. Reddy, Sebastian J. Machera.

**Writing – original draft:** Anay R. Reddy, Sebastian J. Machera.

**Writing – review & editing:** Anay R. Reddy, Sebastian J. Machera, Zoe T. Cook, Engin Özkan, Kang Shen.

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
