## [Decision Letter · Decision Letter 0]

15 May 2025

PGENETICS-D-25-00441

A dendritic guidance receptor functions in both ligand dependent and independent modes

PLOS Genetics

Dear Dr. Shen,

Thank you for submitting your manuscript to PLOS Genetics. After careful consideration, we feel that it has merit but does not fully meet PLOS Genetics's publication criteria as it currently stands. Therefore, we invite you to submit a revised version of the manuscript that addresses the points raised during the review process.

Please submit your revised manuscript within 60 days Jul 14 2025 11:59PM. If you will need more time than this to complete your revisions, please reply to this message or contact the journal office at plosgenetics@plos.org. Please include the following items when submitting your revised manuscript:

We look forward to receiving your revised manuscript.

Kind regards,

Chun Han, Ph.D.

Academic Editor

PLOS Genetics

Fengwei Yu

Section Editor

PLOS Genetics

Aimée Dudley

Editor-in-Chief

PLOS Genetics

Anne Goriely

Editor-in-Chief

PLOS Genetics

**Journal Requirements:**

At this stage, the following Authors/Authors require contributions: Anay R. Reddy, Sebastian J. Machera, Zoe T. Cook, Wioletta I. Nawrocka, Engin Özkan, and Kang Shen. Please ensure that the full contributions of each author are acknowledged in the "Add/Edit/Remove Authors" section of our submission form.

The list of CRediT author contributions may be found here: https://journals.plos.org/plosgenetics/s/authorship#loc-author-contributions

4) Please ensure that the funders and grant numbers match between the Financial Disclosure field and the Funding Information tab in your submission form. Note that the funders must be provided in the same order in both places as well.

**Reviewers' comments:**

Reviewer's Responses to Questions

**Comments to the Authors:**

Reviewer #1: The authors present their work dealing with ligand dependent and independent functions of the cell surface receptor dma-1 in PVD neuron dendrite development. They designed a non-ligand binding receptor version that is able to support dendritic outgrowth by recruiting intracellular binding partners required for actin polymerization. However, ligand binding deficient dma-1 does not support ordered dendritic development, similar to its ligand sax-7 in mutant worms. They further show that dma-1 and ligand binding is required for maintenance of already formed dendritic structures using temperature sensitive or degradable dma-1 alleles. The authors argue that the receptor is sufficient for recruiting intracellular effectors mediating dendritic growth, while structured growth and maintenance requires interaction with the ligand present in other tissues (i.e. hypodermis).

Overall, this is a very neat study showing ligand dependent and independent functions of the dma-1 receptor during development and maintenance of PVD dendrites. While the experiments are well executed and thorough, the interpretation and context with previous studies from this and other systems is very one-sided. I would urge the authors to take a more balanced view and take into account signals that regulate dendrite development differently. What they found for DMA-1 is one of many different flavors of the regulation of dendrite development.

Specific points:

1. The dma-1 non-binding and ligand mutants showed highly disorganized dendrite growth. Can the authors comment on self-avoidance defects as dendrites seem to overlap and possibly adhere to each other extensively in the absence of ligand binding.

2. The authors labeled endogenous sax-7 with mNeonGreen and investigated alignment of dendrites with sax-7 expression (Fig. 3). I have two issues with this data: the authors point out sublateral misalignment of dendrites and sax-7 in controls, particularly of dendrites growing at a non-perpendicular angle. Is that also observed in controls without mNeonGreen-labeled sax-7 or does it indicate a partial functional impairment of sax-7? Second, the authors do not comment on the differences of the sax-7 signal in their example image. Is sax-7 indeed localized to the sublateral domain in dma-1 mutants to a similar extent as in controls? To me it seems there is only weak detectable signal and additionally intracellular accumulations suggesting sax-7 localization partially depends on dma-1 binding. Levels and/or localization should be quantified in controls vs. dma-1 mutants.

3. The authors disregard prior work showing that removal of a ligand or receptor can also lead to the same phenotype and lack of dendritic patterning/growth by a failure of stabilization, e.g. instructive growth: robo/slit: PMID: 17933790, Ten-m: PMID: 24290980, NetB/fra: PMID: 21871804, permissive growth: Ret/mav: PMID: 30157422, 25764303, HSPG/RPTP: PMID: 28874572, growth/maintenance: Celsr PMID: 15296717 etc.). Instead, they refer to work where self-avoidance was analyzed dynamically showing that stochastic growth via repulsion can lead to ordered dendrite morphogenesis. This is similar in regard to stochastic growth yet distinct from the authors’ proposed mechanism presented in this study. The discussion should be more balanced in this regard.

4. The authors’ model suggests that 2ndary branch growth is stochastic and then stabilized by ligand-mediated contact once reaching the sublateral region. However, typically 2ndary branches are oriented perpendicularly to the primary branch in their final configuration, suggesting that somehow the final structure is stabilized and straightened. Their model does not explain how this is achieved. Can sax-7 be released and, e.g., form a gradient or is it purely contact-dependent? For this, some level of live imaging during 2ndary branch growth is needed. The authors should also discuss other possibilities, e.g., previous work showed that mnr-1 regulates dma-1 cell surface levels and kpc-1 cleaves mnr-1 (PMID: 37721334), which might also regulate sax-7 surface levels. This study is not mentioned or discussed.

5. The degradable dma-1 allele (as well as the ts allele at 16°C) seem to be only partially functional based on the authors’ analysis of 4° branches. This should be clearly mentioned as this represents an already sensitized background that is not fully comparable to a wild type situation.

Minor:

1. The authors cite numerous reviews related to their work but little primary references. This should be amended for the main points of their interpretation.

2. The first chapter of results is basically a summary of known findings based on the previous work of the authors and others. Please move this to the introduction.

Reviewer #2: In this manuscript, Reddy et al. identified three critical residues in the extracellular domain of the DMA-1 dendrite guidance receptor, and showed that these mutations abolish the binding between DMA-1 and the dendrite ligand complex in vitro and proper PVD dendrite morphogenesis in vivo. Using confocal imaging, they compared the PVD dendrite morphologies in dma-1 null, ligand null and dma-1 (non-binding) mutant strains, and found that dma-1 (non-binding) mutant animals phenocopy ligand null mutants. In addition, dendrites can grow with or without SAX-7 signals, suggesting that PVD dendrite can grow in a ligand complex-independent manner. Genetic double mutant analyses showed that dendrite outgrowth in the dma-1 (non-binding) mutants relies on HPO-30 and TIAM-1, which are critical regulators for high-ordered branch formation in normal menorah formation as reported previously. The authors further used a temperature-sensitive dma-1 allele and auxin-induced protein degradation assay, and nicely showed that DMA-1 and the receptor-ligand interaction are required for dendrite adhesion/stabilization. Overall, the quality of most results is high, and most of the conclusions are well supported by their data. A few additional experiments may strengthen the quality of this study.

1. In Figure 1, the authors performed a SPR assay to show that a mutant form of DMA-1 ECD (R106D-S107A-R109E) completely abolishes the receptor-ligand interaction in vitro. This is a very nice assay using purified proteins. However, it seems that they cannot perform these experiments using endogenously expressed full length proteins due to technical issues. As each of DMA-1, SAX-7 and MNR-1 contains a single transmembrane domain, it is critical to use alternative assays to test whether the three mutations really abolish the receptor-ligand binding (ideally in vivo). Thus, it would be nice for the authors to test this using the single molecule pull-down and the fly S2 cell aggregation assays, which were successfully used in their previous studies (PMID: 24120131; PMID: 27705746; PMID: 29738713).

2. Did the authors examine whether the three mutations (R106D-S107A-R109E) affect endogenous DMA-1 folding, trafficking or localization? This could be done by tagging both the WT and mutant form of DMA-1 with GFP using CRISPR and then perform confocal imaging.

3. In Figure 2, they imaged PVD dendrite morphologies in both WT and mutant animals, and found that dma-1(non-binding) mutants phenocopy sax-7(null), as they generated more and longer dendrites when compared with the dma-1 null mutant animals. What is the underlying mechanism? This phenotype in dma-1(non-binding) could be explained by either increased outgrowth activity or reduced retraction or both, which can only be tested using time-lapse recording analysis. Although DMA-1 (non-binding) cannot bind with the SAX-7-MNR-1-LECT-2 ligand complex as supported by the SPR assay, it possibly binds to a yet-identified ligand. Thus, it would be nice if they can perform time-lapse analyses for WT and all the receptor/ligand mutants tested here, and add dma-1 delta ECD (reported in PMID: 38766073) as a control.

4. The authors used both a dma-1(ts) allele and an auxin-induced protein degradation assay to test whether DMA-1 is required for branch stabilization, and whether the receptor-ligand binding is required. In Figure 5, the number of tertiary and quaternary branches were quantified. However, in Figure 6, only the number of the quaternary branches were quantified. It seems to this reviewer that the severity of the dma-1 (ts) mutants in 25°C is more severe than the dma-1-gfp-aid animals with auxin. Is that true? If so, why is that? For the results showed in Figure 6, is the receptor-ligand interaction specifically required for the stabilization of quaternary branches, but not for secondary or tertiary ones? Based on the results showed in S1 Fig, it seems that the auxin-induced degradation of endogenous DMA-1 is robust. Is it possible to measure the DMA-1 degradation efficiency using an alternative assay such as western blotting? In addition, could the author explain why dma-1::gfp::aid/+ animals grow more quaternary branches than the dma-1::gfp::aid animals do?

5. To provide additional evidences to demonstrate that DMA-1(R106D-S107A-R109E) specifically affects dendrite stabilization but not outgrowth, the authors may test whether their dma-1(non-binding) mutations affect seam cell trapping induced by ectopically expressed SAX-7 in the seam cells (as reported in PMID: 24120131 and PMID: 24120132).

Reviewer #3: In this manuscript, Reddy and colleagues, investigate the mechanism of how the leucine-rich transmembrane receptor DMA-1 in PVD functions in dendrite morphogenesis. Prior work had established that DMA-1, and its ligands SAX-7, MNR-1, and LECT-2 are required for patterning of PVD somatosensory dendrites. Early on, it was noted that the mutant phenotypes of DMA-1 null mutants were more severe than the ligand mutants suggesting that DMA-1 may serve ligand independent functions. This is observation is the premise for the current manuscript. Using Alphafold, the authors model the structure of the SAX-7/DMA-1 complex and identify three key residues in DMA-1 predicted to constitute key residues for the molecular interaction of the cell adhesion molecule SAX-7 and DMA-1. Biochemical experiments using Surface Plasmon Resonance establish that indeed these three residues are key for the SAX-7/DMA-1 interaction in vitro. Consistent with the hypothesis, the authors further find that introducing these three point mutants into the DMA-1 locus results in phenotypes that are indistinguishable from the ligand mutants SAX-7, MNR-1, and LECT-2 (ie less severe than the DMA-1 null mutants). Additionally, the dendrites of the non-binding DMA-1 allele fails to colocalize with a SAX-7 reporter. Thse findings are consistent with the interpretation that DMA-1 binds to SAX-7 in the same manner in vivo. The authors then go on to show that both ligand-dependent and ligand-independent DMA-1 function through the same downstream mechanisms, based on the observation that mutations in the downstream effectors appear epistatic to the DMA-1 non-binding mutant. The manuscript ends with data suggesting that DMA-1 ligand binding is important for dendrite stability and maintenance. Overall, this is a well written paper that adds an important new facet to the understanding of dendrite patterning. The genetic and biochemical experiments are rigorous and well executed and generally support the conclusions. I have some comments that should be addressed:

1. The biochemical and genetic experiments are consistent with the model the authors propose, namely that ligand-free action of DMA-1 is sufficient for (stochastic) outgrowth and that ligand binding leads to stabilization of the processes. To further support their model, the authors should test whether an endogenous DMA-1 reporter harboring the three point mutations that abolish binding to SAX-7 shows similar staining patterns as a wild type DMA-1 reporter. This experiment would exclude that trafficking or misfolding of the DMA-1 receptor are responsible for the partial loss of function phenotype, a possibility the authors do not entertain.

2. The authors should consistently write the vertebrate homologs with the C elegans gene names to make the manuscript more accessible to a vertebrate readership.

3. Are there potentially analogous findings in the vertebrate literature where receptor knockouts are more severe than ligand knockouts, where similar mechanisms could be at play? If so, that may be worth mentioning to broaden the significance.

4. The authors state that the lect-2; mnr-1; sax-7 triple null animals exhibit a slightly increased number of quaternaries when compared to the sax-7 (null) and dma-1 (non-binding). I don’t see this in the data (only high order branches are shown) or did I miss it? Could the authors clarify and if true, what would the authors make of this observation?

5. Somewhat confusing in the author’s model is the fact that the same downstream signaling mechanisms are employed by ligand dependent and ligand-independent functions of DMA-1? Something must be different between the two. What is it, if not the downstream signaling molecules? Could the authors at least speculate?

6. The structure of MNR-1 is conspicuously absent from the model. Why?

Minor edits:

Line 99: underline after a period.

Line 139: PVD was described essentially concomitantly by three labs: Oren-Suissa et al 2010 (PMID: 20448153), Smith et al. 2010 (PMID: 20537990) and Albeg et al. 2011 (PMID: 20971193). All three papers should be cited here.

Line 183: does the SAX-7 ectodomain bind to DMA-1 in the absence of fused MNR-1?

Line 259: while the authors make a correct conclusion here, both hpo-30 and tiam-1 could also be acting in parallel. This must be mentioned as a possibility.

Line 275: …whether DMA-1 has A role

**Have all data underlying the figures and results presented in the manuscript been provided?**

Reviewer #1: Yes

Reviewer #2: Yes

Reviewer #3: Yes

PLOS authors have the option to publish the peer review history of their article (what does this mean? ). If published, this will include your full peer review and any attached files.

**Do you want your identity to be public for this peer review?** For information about this choice, including consent withdrawal, please see our Privacy Policy .

Reviewer #1: No

Reviewer #2: No

Reviewer #3: No

**Figure resubmission:**
---

## [Decision Letter · Decision Letter 1]

28 Oct 2025

Dear Dr Shen,

We are pleased to inform you that your manuscript entitled "A dendritic guidance receptor functions in both ligand dependent and independent modes" has been editorially accepted for publication in PLOS Genetics. Congratulations!

Yours sincerely,

Chun Han, Ph.D.

Academic Editor

PLOS Genetics

Fengwei Yu

Section Editor

PLOS Genetics

Aimée Dudley

Editor-in-Chief

PLOS Genetics

Anne Goriely

Editor-in-Chief

PLOS Genetics

BlueSky: @plos.bsky.social

Comments from the reviewers (if applicable):

Reviewer's Responses to Questions

**Comments to the Authors:**

Reviewer #1: The authors have adequately addressed all my concerns and I congratulate them on their excellent work.

I have just one comment for their consideration since they emphasize that ligand-independent recruitment of signaling molecules to promote dendrite growth seems so unique. A straightforward interpretation of their data is that they observe autoactivation of the dma-1 receptor without the ligand, as dma-1 levels are elevated in general and redistributed to or close to the cell surface. The lack of endocytosis and increased receptor density might simply recruit more of the adaptors and signaling molecules to the cell surface to induce actin nucleation. Similar effects can be seen if cell surface receptors are overexpressed. e.g. for receptor tyrosine kinases, which typically results in autoactivation.

Reviewer #2: All of my previous concerns have been properly addressed by the authors during revision. Thus, I recommend publication of the revised manuscript.

**Have all data underlying the figures and results presented in the manuscript been provided?**

Reviewer #1: Yes

Reviewer #2: Yes

PLOS authors have the option to publish the peer review history of their article (what does this mean? ). If published, this will include your full peer review and any attached files.

**Do you want your identity to be public for this peer review?** For information about this choice, including consent withdrawal, please see our Privacy Policy .

Reviewer #1: No

Reviewer #2: No

**Data Deposition**

http://datadryad.org/submit?journalID=pgenetics&manu=PGENETICS-D-25-00441R1

**Press Queries**

---

## [Editor Report · Acceptance letter]

PGENETICS-D-25-00441R1

A dendritic guidance receptor functions in both ligand dependent and independent modes

Dear Dr Shen,

We are pleased to inform you that your manuscript entitled "A dendritic guidance receptor functions in both ligand dependent and independent modes" has been formally accepted for publication in PLOS Genetics! Your manuscript is now with our production department and you will be notified of the publication date in due course.

With kind regards,

Aiswarya Satheesan

PLOS Genetics

On behalf of:
